 

# Energetics and conformational pathways of functional rotation in the multidrug transporter AcrB

Yasuhiro Matsunaga[1,2]*, Tsutomu Yamane[3], Tohru Terada[4], Kei Moritsugu[3], Hiroshi Fujisaki[5], Satoshi Murakami[6], Mitsunori Ikeguchi[3], Akinori Kidera[2]

[1]RIKEN Advanced Institute for Computational Science, Kobe, Japan; [2]JST PRESTO, Kawaguchi, Japan; [3]Graduate School of Medical Life Science, Yokohama City University, Yokohama, Japan; [4]Graduate School of Agricultural and Life Sciences, The University of Tokyo, Tokyo, Japan; [5]Department of Physics, Nippon Medical School, Kawasaki, Japan; [6]Graduate School of Bioscience & Biotechnology, Tokyo Institute of Technology, Yokohama, Japan

**Abstract** The multidrug transporter AcrB transports a broad range of drugs out of the cell by means of the proton-motive force. The asymmetric crystal structure of trimeric AcrB suggests a functionally rotating mechanism for drug transport. Despite various supportive forms of evidence from biochemical and simulation studies for this mechanism, the link between the functional rotation and proton translocation across the membrane remains elusive. Here, calculating the minimum free energy pathway of the functional rotation for the complete AcrB trimer, we describe the structural and energetic basis behind the coupling between the functional rotation and the proton translocation at atomic resolution. Free energy calculations show that protonation of Asp408 in the transmembrane portion of the drug-bound protomer drives the functional rotation. The conformational pathway identifies vertical shear motions among several transmembrane helices, which regulate alternate access of water in the transmembrane as well as peristaltic motions that pump drugs in the periplasm.

DOI: https://doi.org/10.7554/eLife.31715.001

*For correspondence:
ymatsunaga@riken.jp

Competing interests: The authors declare that no competing interests exist.

## Introduction

Bacterial multidrug resistance (MDR) is an increasing threat to current antibiotic therapy (*Li et al., 2015*). Resistance nodulation cell division (RND) transporters are one of the main causes of MDR in Gram-negative bacteria. These transporters pump a wide spectrum of antibiotics out of the cell by means of proton- or sodium-motive forces, conferring MDR to the bacterium when overexpressed. Understanding the mechanism of the drug efflux process is invaluable for the treatment of bacterial infections and the design of more effective drugs or inhibitors.

In *Escherichia coli*, the AcrA-AcrB-TolC complex is largely responsible for MDR against many lipophilic antibiotics (*Du et al., 2015*). This tripartite assembly spans the periplasmic space between the inner and the outer membranes of the cell, transporting drugs from the cell to the medium. TolC, an outer membrane protein, forms a generic outer membrane channel in which drugs can passively move towards the medium. AcrB is an inner membrane protein, primarily responsible for specificity towards drugs and their uptake, as well as for energy transduction. AcrA acts as an adaptor that bridges TolC and AcrB (*Du et al., 2014*; *Wang et al., 2017*).

AcrB is one of the best characterized RND transporters in both experiments and simulations, making it a prototype for studying the drug efflux mechanism of the RND family (*Pos, 2009*). AcrB transports drugs from the inner membrane surface or the periplasm to the TolC channel using the

proton-motive force. The structure of AcrB was first solved in a three-fold symmetric form (*Murakami et al., 2002*), and later in an asymmetric form (*Murakami et al., 2006*; *Seeger et al., 2006*; *Sennhauser et al., 2007*). AcrB is a homotrimer with a triangular-prism shape, and each protomer is made up of three domains (*Figure 1A*): the transmembrane (TM) domain and an extensive periplasmic portion comprising the porter and the funnel domains. The TM domain transfers protons across the inner membrane down the electrochemical gradient (*Figure 1C*). The porter domain is made of four subdomains, PN1, PN2, PC1 and PC2 (*Figure 1B*), and is responsible for drug transportation. The funnel domain has a funnel-like shape with an exit pore that indirectly connects to TolC via AcrA (*Du et al., 2014*; *Wang et al., 2017*).

In the asymmetric structure, each protomer of AcrB adopts a distinct conformation that assumes one of the three functional states in the drug transport cycle: Access (*Murakami et al., 2006*) (or Loose [*Pos, 2009*; *Seeger et al., 2006*]), Binding (Tight) and Extrusion (Open) states, or A, B and E, respectively. The A state has entrances (an open cleft between PC1 and PC2, and a groove between TM helices, TM8 and TM9) for drug uptake. The B state has a binding pocket (called the distal binding pocket) formed by PN1, PN2 and PC1 (*Figure 1B*). This pocket contains many hydrophobic residues, as well as several charged and polar residues, contributing to broad substrate 'polyspecificity' (*Takatsuka et al., 2010*; *Vargiu and Nikaido, 2012*). The E state has a drug exit pathway to the exit pore formed by an inclined central helix (*Figure 1B*). The tilt of the central helix is achieved by rigid body motions of a PC2/PN1 repeat (see *Figure 1B*). This motion also closes the PC1/PC2 cleft, leading to contraction of the binding pocket (*Pos, 2009*).

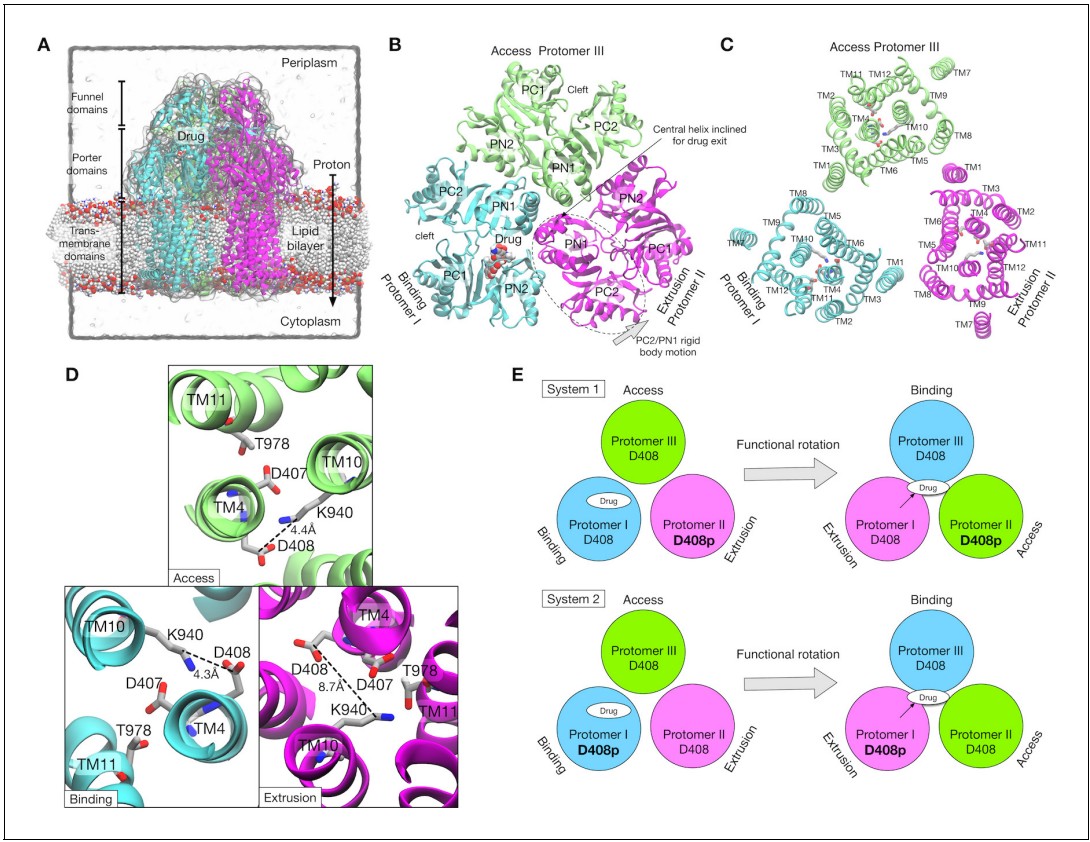

**Figure 1.** Crystal structure of AcrB and simulation setup. (A) Simulation box and a side view of AcrB embedded in a palmitoyloleoyl phosphatidylethanolamine (POPE) lipid bilayer. A drug (minocycline) is represented by spheres. (B) Porter domains of the crystal structure viewed from the cell exterior. The protomers in cyan, magenta and lime represent the Binding, Extrusion, and Access states, respectively. (C) The transmembrane domain viewed from the cell exterior. Key residues that are related to proton translocation are represented by sticks. (D) Close-up view of the key residues. (E) Schematics of the simulation systems. The protonated D408 is denoted by D408p. System 1 has D408p in the Protomer II. System 2 has D408p in the Protomer I.

DOI: https://doi.org/10.7554/eLife.31715.002

These findings from the asymmetric structure and other biochemical data led to the proposal of the functional rotation mechanism (*Murakami et al., 2006*; *Seeger et al., 2006*). Hereafter, following *Yao et al. (2010)*, we use a three letter notation to represent the state of the trimer. For example, the BEA state represents a trimeric state in which the protomer I is in the B state, the protomer II in the E state, and the protomer III in the A state (where protomers are numbered in a counterclockwise order). In the functional rotation mechanism, for example starting from the BEA state, drug efflux is coupled with a conformational transition to the EAB state. Viewed from the top, this transition is a 120 degree rotation of functional states (not a physical rotation). This mechanism is supported by both biochemical studies (*Seeger et al., 2008*; *Takatsuka and Nikaido, 2007*) and molecular dynamics (MD) simulation studies (*Schulz et al., 2010*; *Yao et al., 2010*; *Zuo et al., 2015*).

The three functional states satisfy the conditions required in the alternate access model (*Jardetzky, 1966*): the A state (and possibly also the B state) corresponds to an inward-facing state, the E state represents an outward-facing state, and the B state is an occluded state. However, despite this excellent correspondence between the structures and states, the dynamical picture of functional rotation remains unclear. Specifically, how are the conformational changes (functional rotation) and the energy source (proton translocation) synchronized and coupled dynamically? In contrast to drug efflux through small transporters, where substrate efflux and energy transduction are spatially coupled in the TM domain (*Drew and Boudker, 2016*), drug efflux by AcrB occurs in the periplasmic space, which is spatially separate (~50 Å apart) from the proton translocation sites in the TM domain. To understand the whole process of drug efflux of AcrB, the energy transduction mechanism between the two distant sites should be clarified, which is the purpose of this study.

The TM domain of each protomer contains twelve α-helices (TM1–12, *Figure 1C*). Essential residues for proton translocation are in the middle of the TM helices, D407 and D408 on TM4, K940 on TM10, and T978 on TM11. AcrB mutants in which these residues are substituted by alanine are inactive (*Guan and Nakae, 2001*; *Takatsuka and Nikaido, 2006*). In the asymmetric structure, K940 shows a distinct side chain orientation in the E state compared with that in the A and B states (*Figure 1D*). Specifically, K940 tilts away from D408 and towards D407 and T978 in the E state, suggesting that transient protonation of D408 contributes to the TM conformational change. Recently, all-atom MD simulation by *Yamane et al. (2013)* showed that the protonation of D408 stabilized the E state, whereas deprotonated D408 induced the conformation change of the TM domain toward the A state. *Eicher et al., (2014)* showed, by X-ray crystallography of AcrB mutants combined with MD simulations, that these conformational changes in the TM domain were related to the alternate access of water, and in turn to the functional rotation.

Although these studies provide invaluable insights into the energy transduction mechanism, the difficulty in experimental observation of the protonation states and the time-scale limitation in all-atom MD simulations preclude direct characterization of the energy transduction process. The most direct characterization would be the observation of the functional rotation in various protonation states. To achieve this, we conducted extensive MD simulations with state-of-the-art supercomputing. Using an explicit solvent/lipid all-atom model (480,074 atoms), we identified conformational transition pathways in one step of the functional rotation for the complete AcrB trimers with different protonation states.

Here, the physically most probable pathway for the functional rotation was searched with the string method (*Maragliano et al., 2006*; *Pan et al., 2008*). In this method, instead of running a long MD simulation, a pathway connecting two known structures is optimized using multiple copies of a simulation system. The pathway is represented in an important subspace (called 'collective variables') with discretized beads (called 'images'). The optimization is carried out by updating the coordinates of the images through multiple simulations toward the minimum free energy pathway. To date, the string method has been successfully applied for finding the conformational transition pathways of various protein systems, such as c-Src kinase (*Gan et al., 2009*), myosin VI (*Ovchinnikov et al., 2011*), ion channels (*Lev et al., 2017*; *Zhu and Hummer, 2010*), adenylate kinase (*Matsunaga et al., 2012*), β2-microglobulin (*Stober and Abrams, 2012*), $V_1$-ATPase (*Singharoy et al., 2017*), calcium pump (*Das et al., 2017*), and membrane transporters (*Moradi et al., 2015*; *Moradi and Tajkhorshid, 2014*). Extensive samplings around the images are necessary for accurate estimates of the energetics or free energy changes of the whole system along the pathway (*Matsunaga et al., 2012*; *Moradi and Tajkhorshid, 2014*), where free energy changes

were evaluated by performing umbrella samplings around the images. In addition, to evaluate the impacts of protonation and drug binding, we conducted alchemical free energy calculations (*Klimovich et al., 2015*; *Chipot, 2014*).

By conducting these calculations, we compared the conformational pathways and energetics of two simulation systems, each with different protonation state, to clarify the causal relationship between the protonation and the functional rotation. Free energy evaluation along the pathways shows that the protonation of D408 in the B state induces the functional rotation. Structural analysis along the conformational pathway reveals that vertical shear motions in specific TM helices regulate the alternate access of water in the TM domain, as well as the peristaltic motions pumping a drug in the porter domain. These findings provide a simple and unified view of energy transduction in AcrB.

## Results

### Energetics of functional rotation under two different protonation states

We computed the most probable conformational pathway in the functional rotation from the BEA state to the EAB state with the string method. From the observation of the crystal structure, we postulated that the protonation of D408 in the B state induces the conformational change toward the E state. This hypothesis was tested with the setup of two different simulation systems (*Figure 1E*): System 1 has a protonated D408 (hereafter, denoted by D408p) in protomer II, which undergoes a transition from the E to the A state. System 2 has D408p in protomer I, undergoing transition from the B to the E state. If the hypothesis is correct, then system 1 should be stable at the initial BEA state, whereas the system 2 would trigger the functional rotation. A drug (minocycline) was explicitly included in both systems. It is bound in protomer I (the B state) at first, and then transported to the exit pore during the functional rotation. No additional drug for uptake was included in order to keep the drug–AcrB interaction as simple as possible. For the string method calculation, we first generated an initial pathway in a targeted MD simulation (see Materials and methods), in which the structure of AcrB was guided from the BEA state to the EBA state by means of steering forces, simultaneously with the drug being directed toward the exit pore. The collective variables (Cartesian coordinates of Cα atoms in the porter domain and TM helices, see Materials and methods and *Figure 2—figure supplement 1*) used in the string method were carefully chosen according to previous simulation studies (*Yamane et al., 2013*; *Yao et al., 2010*). The drug was not included in the collective variables because it is diffusive (*Schulz et al., 2010*). Thirty images were used to represent the pathway.

*Figure 2A* shows the free energy profiles along the pathways obtained by the string method, illustrating the convergence towards the minimum free energy pathway. The impact of protonation is revealed when the converged pathways of the two systems are compared (see *Figure 2B*). While system 1 has an energy minimum close to the initial BEA state (at image 5), the minimum of system 2 is shifted towards the final EAB state (at image 15), indicating that the protonation state in system 2 drives the functional rotation. The structure and protonation state of the final EAB state of system 2 is identical to that of the initial BEA state of system 1, when the molecule is rotated by 120 degrees and the bound/unbound drug is ignored. This relation suggests that the release of the drug is the main factor for the increase in free energy in the late stage (images 20–30) of system 2. The molecule may need a new drug molecule to progress to the next step of rotation. Previous simulation studies (*Yamane et al., 2013*; *Yao et al., 2010*) suggested that asymmetric trimeric states are unstable without a bound drug, while the symmetric AAA state becomes most stable, avoiding the rebinding of the released drug.

We quantified the contribution from the drug–protein interactions at each image by calculating the absolute drug binding free energy to AcrB (see Materials and methods). The absolute binding free energy in system 2 is plotted in *Figure 2B* with the black line. The binding free energy increases rapidly at around images 10–20, prompting detailed structural analyses of the implication (*Figure 2C and D*). The closing of the binding pocket was monitored by the distance between the side chains of F178 and F615 (*Eicher et al., 2012*), and the exit gate opening by the Q124 to Y758 distance (*Schulz et al., 2010*), as well as the position of the drug (*Figure 2C*). In images 5–15, the binding pocket collapses and the exit gate opens, indicating that the minimum free energy state

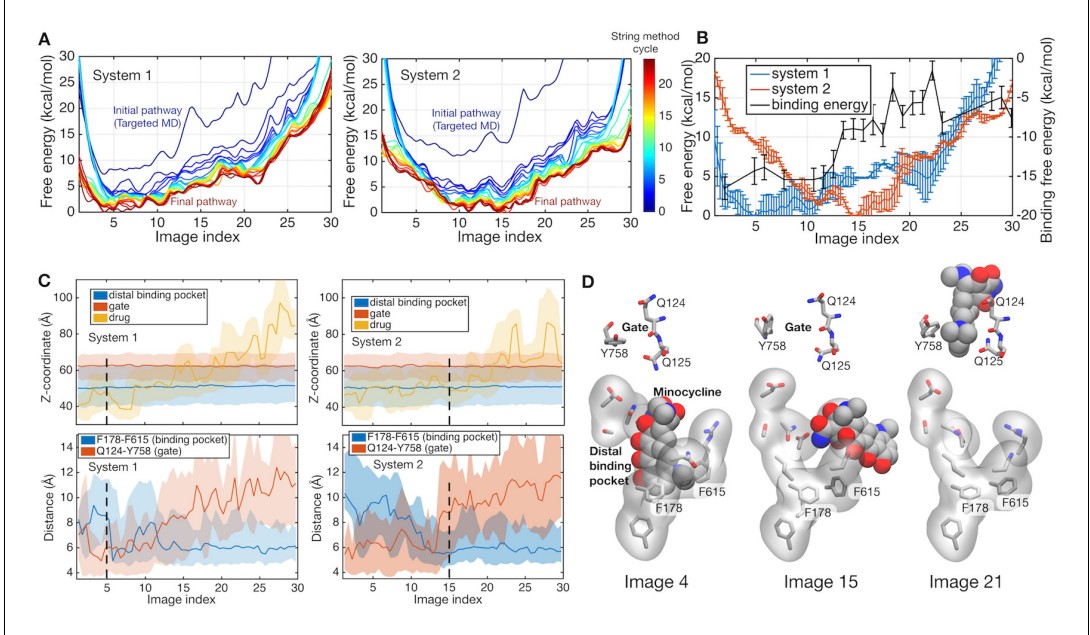

**Figure 2.** Energetics of functional rotation for two different protonation states. (**A**) Free energies of the whole system along the conformational pathways obtained by the string method for systems 1 and 2. Image 1 represents the BEA state, whereas image 30 corresponds to the EAB state (see text for the three-letter notation). Line color represents the cycle of the string method calculation. The initial pathway (obtained by the targeted MD) is indicated by a dark blue line, whereas the final converged pathway is in dark brown. (**B**) Comparison of the final converged pathways for systems 1 (blue) and 2 (red). Free energies are referenced to their minima (against the left *y*-axis). The black line indicates the absolute binding free energy of the drug for each image of system 2 (against the right *y*-axis). Statistical uncertainties are represented by error bars. (**C**) *Z*-coordinate (perpendicular to the membrane) of binding pocket residues, gate residues, and drug atoms, and distances between F178 and F615 and between Q124 and Y758 are shown. Light color regions represent minimum and maximum ranges, and dark lines the average. The positions of minima in system 1 and 2 are indicated by black broken lines. (**D**) Representative structures of the binding pocket and gate residues (sticks) and of the drug (spheres) are shown at images 4, 15 and 21.

DOI: https://doi.org/10.7554/eLife.31715.003

The following figure supplements are available for figure 2:

**Figure supplement 1.** Collective variables used for the string method.
DOI: https://doi.org/10.7554/eLife.31715.004

**Figure supplement 2.** Relaxation of structures before string method calculation.
DOI: https://doi.org/10.7554/eLife.31715.005

**Figure supplement 3.** Root mean square displacements (RMSDs) of images from those of the initial pathway.
DOI: https://doi.org/10.7554/eLife.31715.006

**Figure supplement 4.** Comparison of string method pathways and brute-force simulations.
DOI: https://doi.org/10.7554/eLife.31715.007

(image 15 in system 2) is ready to release the drug. In system 2, the initial loss of the binding free energy is compensated for by the impact of the protonation on the functional rotation (as discussed in the next subsection). In images 10–20, the drug diffuses into the gate weakly interacting with the tunnel (*Figure 2C*). As discussed by *Schulz et al. (2010)* in their targeted MD simulation study, this process may be diffusion-limited. In summary, system 2 represents the protonation state that promotes drug extrusion mediated by the functional rotation.

## Protonation induces a vertical shear motion in the TM domain

How does the protonation at D408 induce the conformational change in the TM domain? To address this question, we first investigated the impact of the protonation on the minimum free energy state (image 5) of system 1 by alchemically transforming it towards system 2, that is, by protonating D408 of protomer I and deprotonating D408p of protomer II (see Materials and methods). Electrostatic potential energy maps before and after the transformation of the protonation state reveal that the TM domain of protomer I is destabilized by the protonation because of a strong positive

electrostatic potential bias (*Figure 3A and B*). This bias is caused by the surrounding positively charged residues R971 and K940, whose side-chains orient towards D408p. The relaxation of the potential bias can be seen when inspecting image 15 of system 2, the minimum free energy state (*Figure 3C* and *Figure 3—figure supplement 1*). The side-chain of R971 (on TM11) reorients downwards (towards the cytoplasm), as does D408p (on TM4) due to repulsion from K940 (on TM10).

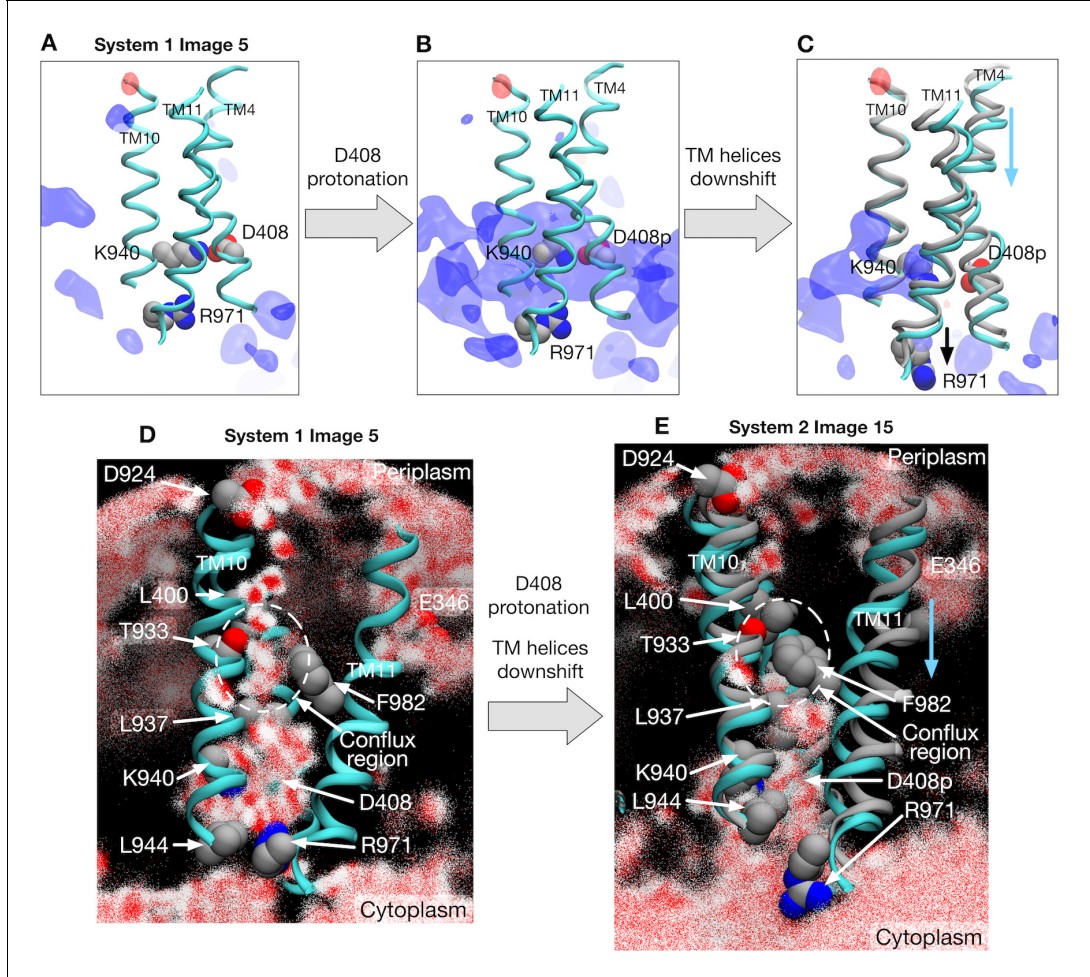

**Figure 3.** Electrostatic features and water molecule distributions in the transmembrane region. (**A**) Representative structure of transmembrane helices 4, 10 and 11 for protomer I (cyan) of system 1 at image 5 drawn with cartoons. Deprotonated/protonated D408, K940 and R971 are represented by spheres. Averaged electrostatic potential isosurfaces are drawn in blue (corresponding to the isovalue of 0.06 kcal/mol) and red (−0.06 kcal/mol). (**B**) After transforming the protonation state toward system 2. (**C**) System 2 at image 15. For comparison, the helices of image 5 are drawn in gray. (**D**) 2500 snapshots of water atoms are drawn with red points (oxygen) and white points (hydrogen) for protomer I of system 1 at image 5. Key residues (L400, D924, T933, L937, K940, F982 and R971) are drawn with spheres. (**E**) System 2 at image 15. For comparison, the helices of image 5 are drawn in gray.
DOI: https://doi.org/10.7554/eLife.31715.008

The following figure supplements are available for figure 3:

**Figure supplement 1.** Structural changes and electrostatic features in the transmembrane region.
DOI: https://doi.org/10.7554/eLife.31715.009

**Figure supplement 2.** Electrostatic features with protonated D924 and E346.
DOI: https://doi.org/10.7554/eLife.31715.010

**Figure supplement 3.** Downshift and upshift motions of transmembrane helices.
DOI: https://doi.org/10.7554/eLife.31715.011

**Figure supplement 4.** Comparison of transmembrane helix positions in the crystal structure.
DOI: https://doi.org/10.7554/eLife.31715.012

**Figure supplement 5.** Water molecule distributions in transmembrane domain.
DOI: https://doi.org/10.7554/eLife.31715.013

These motions involve the downshift of TM4 and TM11, whereas TM10 remains at the original position. In order to quantify this relaxation, free energy differences over the alchemical transformation (from system 1 to system 2) were evaluated at images 5 and 15 (see Materials and methods), which yielded $25.9 \pm 0.5$ kcal/mol at image 5, corroborating electrostatic destabilization of the protonation state in system 2, and only $0.4 \pm 6.8$ kcal/mol at image 15, suggesting that the protonation no longer affects the energetics at this stage.

We examined another possible protonation scheme. Here, D924 and E346, located at the periplasmic side of the TM domain, were selected for the protonation sites according to the pKa calculation by *Eicher et al. (2014)*. The electrostatic potential map after the protonation shows a local positive change around D924, which may stabilize the system through interactions with the oxygens of the phosphate groups of lipids, and which may help to recruit water molecules for proton uptake (see the next subsection). However, the impact of the protonation at D924 is spatially localized compared to that of D408, and the change in the electrostatic potential was not propagated to the transmembrane region. The protonation at E346 scarcely changed the electrostatic potential. Therefore, these protonation sites may not have a strong impact on the functional rotation.

By monitoring the centers-of-mass for the other TM helices along the conformational pathway, we found that downshifts occur not only in TM4 and TM11 but also in TM3, TM5 and TM6 (*Figure 3—figure supplement 3*). Based on their analysis of crystal structures, *Eicher et al. (2014)* recognized two rigid assemblies in the TM domain, R1 (TM1 and TM3 to TM6) and R2 (TM7 and TM9 to TM12), which change their mutual arrangement during the transitions among B, E and A states. In terms of R1/R2 rigid body motions, the downshift motions of the TM helices can be interpreted as a vertical shear motion of R1 against R2. Tight packing of the helices within R1, and rather weak interactions on the interface between R1 and R2 may facilitate such a large rearrangement. TM11 in R2 is the exception, because it moves downwards together with R1. Previous structural studies mostly focused on lateral shear motions and rocking motions (*Du et al., 2015*; *Eicher et al., 2014*) of R1 and R2, whereas our observations clarify that protonation mainly induces a vertical shear motion. Indeed, downshifts of the specific TM helices, including TM11, can be discerned in the crystal structures when the TM helices of different states are properly aligned (*Figure 3—figure supplement 4*).

## Vertical shear motion in TM domain regulates the alternate access of water

The next question is how D408 becomes protonated. Because proton translocation across the TM domain requires the access of water molecules to supply protons, we investigated the distribution of water molecules in protomer 1 (*Figure 3D and E*, and *Figure 3—figure supplement 5* for other protomers). At image 5 of system 1 with D408, a water wire channel is clearly observed from periplasmic D924 (a possible transient proton binding site to facilitate proton uptake on TM10 [*Eicher et al., 2014*]) to the space in the middle of the helices (called the conflux space [*Fischer and Kandt, 2011*]) surrounded by hydrophobic residues (*Figure 3D*). Also, a minor water pathway comes from E346 (another possible transient proton-binding site on TM2), which merges into the conflux space. At the cytoplasmic end, the side chain of R971 breaks the channel, preventing the leakage of protons (*Eicher et al., 2014*).

By contrast, water distribution at image 15 of system 2 with D408p is completely different due to the vertical shear motion in the TM domain (*Figure 3E*). There is no water channel coming from the periplasm, resulting from the collapse of the conflux space by the downshifts of TM4 and TM11; distances between residue pairs L400–T933 and L937–F982 decrease, preventing water molecules from entering the conflux space (*Figure 3E*). On the other hand, the side chain of R971 opens towards cytoplasmic bulk water, exposing the protonation site and providing a possible proton release pathway. Thus, R971 is an electrostatic gate for proton diffusion to the cytoplasm, analogous to the selectivity filter of aquaporins as discussed in previous studies (*Eicher et al., 2014*; *Eriksson et al., 2013*). In summary, these two states, image 5 of system 1 and image 15 of system 2, exemplify the alternate access of water that occurs concurrently with the proton translocation.

## Vertical shear motion in TM domain is coupled to porter domain motion

Next, we studied energy transduction from the TM domain to the porter domain. Does D408p in the B state dynamically induce the conformational change of the porter domain? *Figure 4* shows the

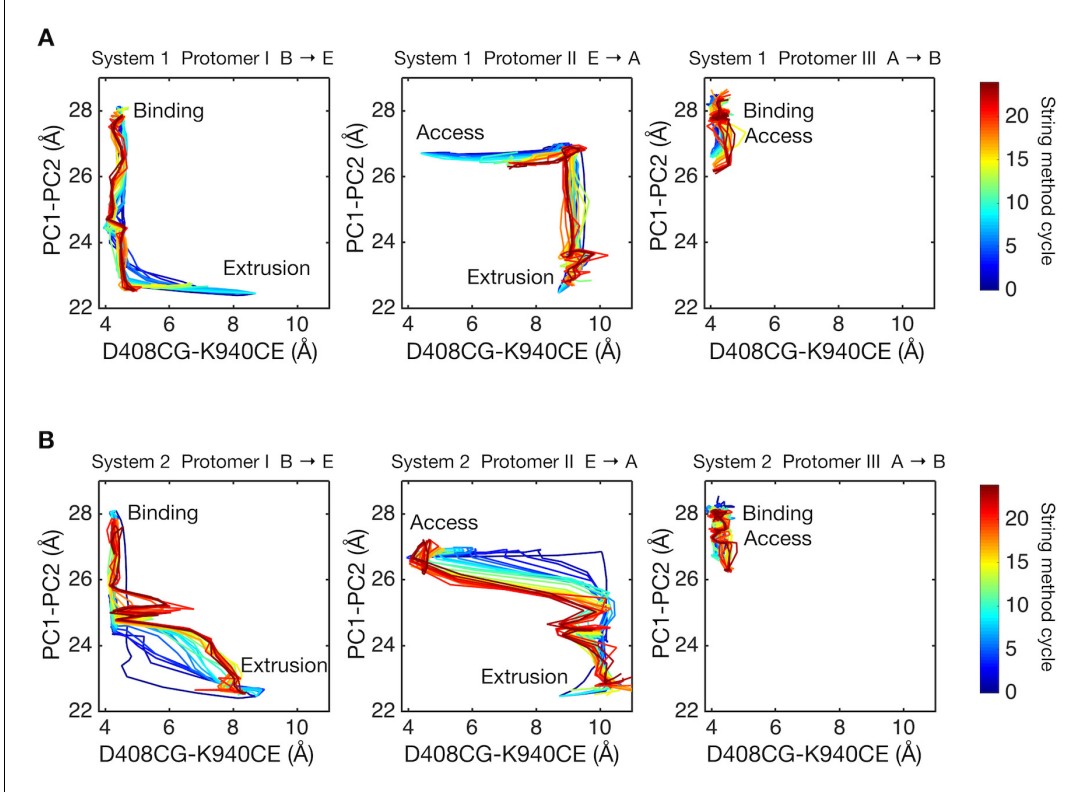

**Figure 4.** Two-dimensional visualization of conformational pathways. (**A**) Conformational pathways of system 1 are monitored by distance between the Cγ-atom of D408 and the Cε-atom of K940 (illustrated in *Figure 1D*), and by the center-of-mass distance between the PC1 and the PC2 subdomains. These quantities are deterministic because they are involved in the collective variables used with the string method calculation. Dark blue lines indicate the initial pathway generated by the targeted MD simulations. Dark brown lines are the final converged pathways obtained by the string method. (**B**) Conformational pathways of system 2.

DOI: https://doi.org/10.7554/eLife.31715.014

conformational pathways of systems 1 and 2 monitored by the distances between D408 and K940 in the TM domain (see *Figure 1D* for crystal structure), and between PC1 and PC2 subdomains in the porter domain (see *Figure 1B*). As mentioned in the Introduction, the closing rigid body motion of the PC2 subdomain is related to the opening of the drug exit pathway, as well as to the contraction of the binding pocket (*Figure 1B* and *Figure 5*). The dark blue lines indicate the initial pathways generated by the targeted MD simulation. As shown in previous studies, targeted MD simulation tends to yield pathways in which local (in our case, TM domain) and global (porter domain) motions are not correlated with each other (*Ovchinnikov and Karplus, 2012*). Remarkably, however, the string method yielded synchronized pathways for system 2, but not for system 1. This suggests the existence of a coupling between the remote sites (the porter and TM domains) in system 2.

How are the two sites coupled? To quantify the dynamic correlations of residue pairs, we employed the mutual information analysis, which has been successfully used in analyses of allosteric coupling compared with experiments (*McClendon et al., 2009*). Mutual information was calculated from snapshots along the minimum free energy pathways obtained with the string method. *Figure 5—figure supplement 1A* plots the number of residue pairs between the TM helices and the porter domain as a function of 'correlation' measured by mutual information. Clearly, system 2 (broken lines in the figure) exhibits larger correlations than does system 1 (solid lines). Protomer I, among the three protomers, shows the greatest correlation, indicating that the TM domain motion of protomer I that is induced by proton translocation is harnessed for the rearrangement of the porter domain. This means that the transition from the B to the E state is the energy-consuming step (*Pos, 2009*).

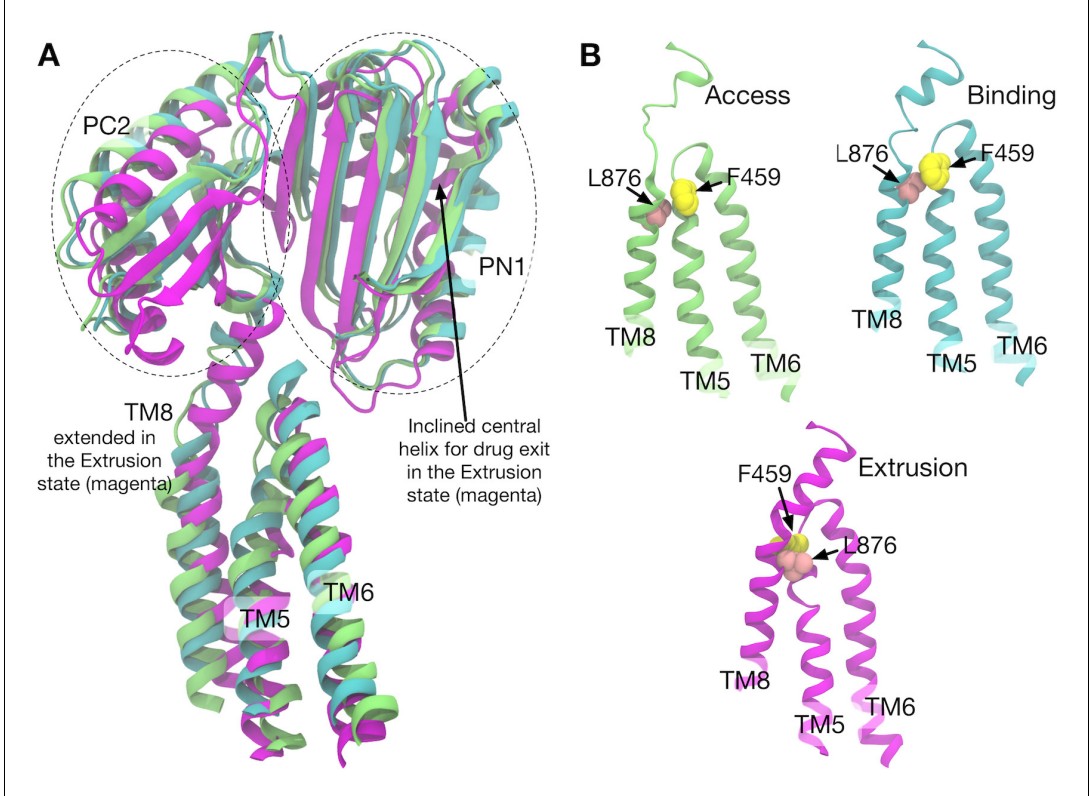

**Figure 5.** Comparison of the structures of transmembrane helices 8, 5, 6 and porter subdomains PC2 and PN1 in the crystal structure. (A) Side view of TM8, TM5, TM6 and porter subdomains PC2 and PN1 in the Binding (cyan), Extrusion (magenta) and Access (lime) states, in which the structures are superimposed. (B) Relative positions of F459 (yellow) on TM5 and L876 (pink) on TM8 are compared in different states.
DOI: https://doi.org/10.7554/eLife.31715.015

The following figure supplements are available for figure 5:

**Figure supplement 1.** Correlations between the transmembrane helices and the porter domain motions.
DOI: https://doi.org/10.7554/eLife.31715.016

**Figure supplement 2.** Comparison between systems 1 and 2.
DOI: https://doi.org/10.7554/eLife.31715.017

Correlated residue pairs of system 2 are visualized in *Figure 5—figure supplement 1B* (*Figure 5—figure supplement 2* for both systems). Here, residue pairs whose mutual information is larger than a threshold are plotted. In protomer I (B→E), high correlations were observed between TM4–TM6, TM8 and TM11, and PN1 and PC2 (and also weakly with PN2). As identified above, these TM helices (except for TM8) are involved in the downshift motions induced by D408p. To see how the TM helices and the porter domain are correlated, we visualized the structures along the pathway (*Figure 5—figure supplement 1C*). In particular, the coil-to-helix transition of TM8, which leverages the rigid body motion in PN1/PC2 (see *Figure 1B* and *Figure 5*), is sterically hindered by the side-chain of F459 on TM5 in the B state (*Figure 5* and *Figure 5—figure supplement 1C*). Downshift of TM5 causes the F459 sidechain to flip, allowing the coil-to-helix transition of TM8. Thus, the downshift of TM5 reduces the barrier for the extension of TM8, expediting the PN1/PC2 rigid body motion to open the exit gate for the drug.

Protomer I also exhibits weak linkage between PN2 and several TM helices. These weak correlations may arise from PN2's rearrangements relative to PC1 due to the shrinkage of the distal binding pocket (see *Figure 1B* for the binding pocket location). The downshift of the TM helices decreases contacts between the TM domain and PN2, and may accommodate the rearrangement of PN2. Previous studies indicated that a slight tilting of TM2 is coupled to PN2 motion (*Du et al., 2015*; *Eicher et al., 2014*), which is also observed here as weak correlations between TM2 and PN2.

Unlike protomer I, protomer II (E→A) has only a local correlation network between TM5 and TM8 (and weakly with TM10) and PN1 and PC2. This relatively weak correlation is associated with the reverse helix-to-coil transition of TM8, which closes the drug exit gate. Also, weak correlations can be seen in TM1, TM4, and TM10, which are consistent with the scenario described by *Yamane et al. (2013)* for the E→A transition: twisting motions of TM4 and TM10 interfere with TM1, PN1, and PN2. Protomer III (A→B) has rather weak correlations between PN2 and the TM helices, which may be related to the rearrangement of PN2/PC1 on distal binding pocket formation and the influence of rising TM helices on contact with the PN2 domain. These local and weak correlations in protomers II and III suggest that the minor coupling either occurs indirectly via interfacial interactions between protomers or is mediated by drug binding, not being harnessed by proton translocation.

## Discussion

In the light of our results and those of previous studies (*Eicher et al., 2014*; *Seeger et al., 2008*; *Takatsuka and Nikaido, 2007*; *Yamane et al., 2013*; *Yao et al., 2010*), we propose the cycle of the functional rotation summarized in *Figure 6*. Coupling of the TM and the porter domains during B→E is the energy-consuming process. Protonation of D408 in the B state drives a vertical shear motion in the TM domain, which regulates the alternate access of water (the conflux space closes, R971 opens towards the cytoplasm) as well as the motion of the porter domain. In this process, the TM8 works as a transmitter. It's extension induces the rigid body motion of the PN2/PC1 tandem, opening the drug exit gate and shrinking the distal binding pocket. In the E→A step, the porter domain of E relaxes towards A without systematic couplings with the TM domain, perhaps driven mainly by cooperativity (interfacial interactions) between protomers. Cross-linking experiments (*Seeger et al., 2008*; *Takatsuka and Nikaido, 2007*) and coarse-grained model simulations (*Yao et al., 2010*) suggested that steric clashes occur between the E protomers, making two or more E-state protomers in

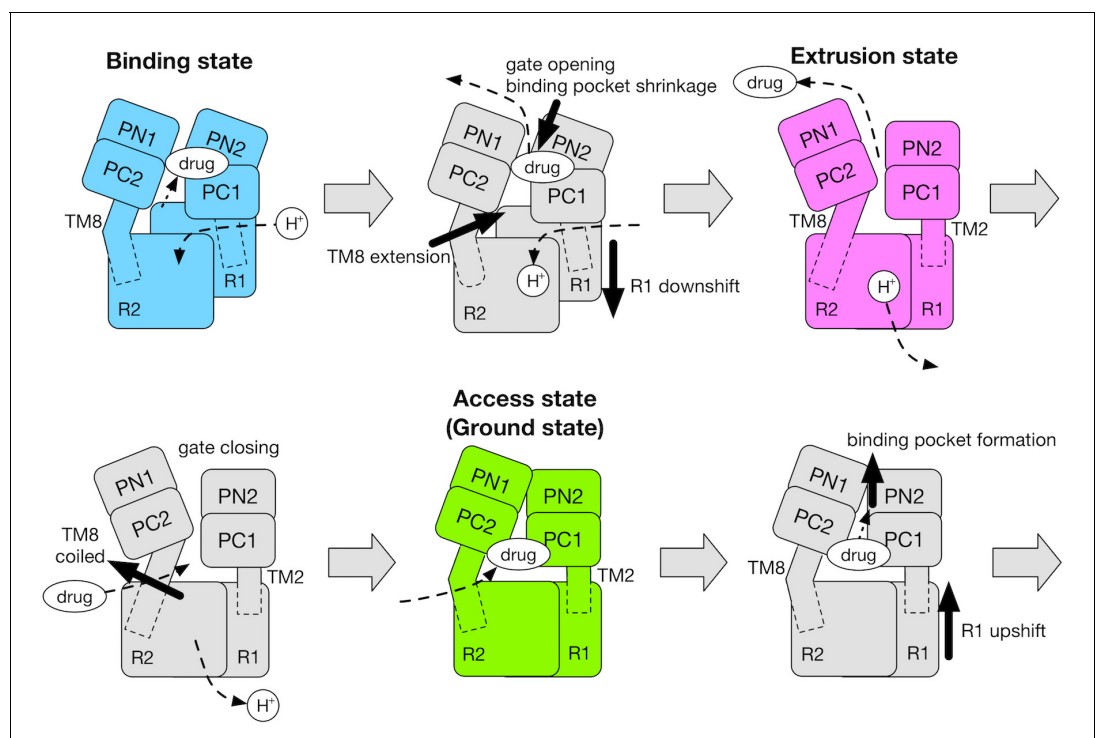

**Figure 6.** Schematic for the mechanism of the functional rotation. Relative motions of the transmembrane domain (consisting of R1 and R2 repeats, and TM2 and TM8) and the porter domain (consisting of the PN1, PN2, PC1 and PC2 subdomains) are represented by thick black arrows. The coil-to-helix and reverse helix-to-coil transitions in TM8 are also indicated by thick black arrows. The accessibility of protons/drug to the TM/porter domain is indicated by small black arrows with broken lines. The funnel domain is omitted for visual clarity.
DOI: https://doi.org/10.7554/eLife.31715.018

the trimer energetically unfavorable, and making the E→A transition occur spontaneously. In the A→B step, drug that is bound in A may 'catalyze' (*Moradi et al., 2015*) formation of the distal binding pocket in the PN2/PC1 tandem. The resulting conformational change increases the number of contacts with the TM domain and drives the TM helices upwards.

This scheme is consistent with the model proposed by *Pos (2009)*, which is based on a combination of the alternate access model and the binding change mechanism for $F_0F_1$-ATP synthase (*Boyer, 1993*). The present study provides evidence that energy is consumed to release the drug (i.e., the B-to-E step). This energy consumption was predicted by Pos's model in analogy to $F_0F_1$-ATP synthase, where the energy is used to release the ATP from the beta-subunit. This study also provides a simple molecular mechanism for energy transduction from the TM domain to the porter domain in terms of the vertical shear motion in the TM domain. The same type of vertical shear motions can be seen in other transporters that employ the so-called elevator alternating-access mechanism (*Drew and Boudker, 2016*; *Slotboom, 2014*).

Finally, we discuss the possibility of other protonation schemes that may impact on functional rotation. The computational work by *Eicher et al. (2014)* indicated that not only D408 but also D407 is protonated in the E state, suggesting a stoichiometry of two protons per one substrate. On the other hand, pKa calculations by *Yamane et al. (2013)* showed that only D408 should be protonated in the E state. Furthermore, their MD simulations showed that the protonation of both D407 and D408 destabilizes the structure of the TM domain. In this study, we have double-checked our results by examining the free-energy profiles along the pathway and alchemical free energies. The consistency of two independent calculations has confirmed that the impact of transient deprotonation/protonation of D408 is certainly enough to drive functional rotation and the extrusion of a drug.

## Materials and methods

### System setup

The crystal structure of the asymmetric AcrB trimer (PDB entry: 4DX5 [*Eicher et al., 2012*]) with bound minocycline in the distal binding pocket of protomer I (B state) was used for MD simulations. The AcrB trimer was embedded in an equilibrated lipid bilayer membrane of POPE (1-palmitoyl-2-oleoyl-*sn*-glycero-3-phosphoethanolamine). We estimated the TM region using TMHMM (*Sonnhammer et al., 1998*) and overlaid it on the equilibrated POPE membrane, then removed lipid molecules within 1.5 Å of the protein. A large hole with a diameter of ~30 Å in the TM center of AcrB should be filled with phospholipids to avoid proton leakage across the membrane. Twelve POPE molecules were manually packed into the hole (six for the upper leaflet and six for the lower leaflet). Then, the simulation box was filled with water molecules. D408 of protomer II (E state) was protonated for system 1 while D408 of protomer I (B state) was protonated for system 2. Finally, sodium ions were added to make the net charge of the system neutral. The final system comprised 480,074 atoms.

Equilibration was performed with MARBLE (*Ikeguchi, 2004*) and mu2lib (publicly available at http://www.mu2lib.org) using CHARMM36 for proteins (*Best et al., 2012*) and lipids (*Klauda et al., 2010*) as force-field parameters. For minocycline, the parameters compatible with CHARMM force-fields, developed by *Aleksandrov and Simonson (2009)*, were used. Electrostatic interactions were calculated with PME (*Darden et al., 1993*), and the Lennard-Jones interactions were treated with the force switch algorithm (*Steinbach and Brooks, 1994*) (over 8–10 Å). The symplectic integrator for rigid bodies (*Ikeguchi, 2004*) and SHAKE (*Ryckaert et al., 1977*) constraint was used with MARBLE and mu2lib, respectively, using a time step of 2 fs. After an initial energy minimization, the system was gradually heated to 300 K for 1 ns. Then, the system was equilibrated for 8 ns under NPT ensemble (300 K and 1 atm). Throughout the minimization and equilibration, positional harmonic restraints were imposed to the nonhydrogen atoms of AcrB. For the minimization and first 2 ns equilibration, positional restraints were also applied to the nonhydrogen atoms of minocycline and crystallographic water molecules.

### String method

To generate the initial pathways to be used in the string method, targeted MD simulations were performed with MARBLE. Starting from the BEA trimeric state, a targeted MD simulation of 10 ns was

conducted towards the EAB state. A force constant per atom of 1.0 kcal/Å$^2$ was used for the harmonic restraint on the root-mean-square displacement (RMSD) variable measured with the nonhydrogen atoms of AcrB. The target structure (EAB state) was created by rotating the initial structure by 120 degrees around the z-axis (perpendicular to the membrane plane). As shown by *Schulz et al. (2010)*, drug efflux towards the exit pore in the targeted MD when pulling only the atoms of AcrB was not observed in 10 ns because the drug extrusion process is diffusion limited. Thus, in order to extrude minocycline toward the exit pore, minocycline was also pulled in our simulation. Starting from the bound form in the distal binding pocket, (the RMSD variable of) minocycline was pulled to 5 Å above the exit gate (Y758) for 10 ns using the same force constant as that of AcrB. The same protocol was applied to both system 1 and 2, yielding very similar trajectories. The trajectories were post-processed for the string method calculation: snapshots were interpolated by piece-wise linear fitting in the collective variable space and 30 equidistant points (called images) were defined as the initial pathway. Image 1 corresponds to the BEA state, and image 30 represents the EAB state. Then, the all-atom coordinates (and corresponding velocities and box size) closest to each image were chosen for the initial structure of the string method.

Collective variables for the string method were carefully chosen according to previous simulation studies. We chose Cartesian coordinates of Cα atoms in the porter domain (PN1, PN2, PC1 and PC2) and selected TM helices (TM4, TM5, TM6, TM8, TM10 and a loop connecting TM5 and TM6) of all protomers (see *Figure 2—figure supplement 1*). The chosen porter domain residues were those used in a previous coarse-grained study, which showed that they successfully capture the essence (thermodynamic and dynamic properties) of drug transportation (*Yao et al., 2010*). The TM helices were chosen because a previous all-atom simulation suggested that those particular TM helices are highly mobile (*Yamane et al., 2013*). According to recent studies, which investigated the impact of collective variable choice on pathway accuracy, it is important to choose such mobile domains (*Matsunaga et al., 2016*; *Pan et al., 2014*). We also chose the Cγ-atoms of D407 and D408 and the Cε-atom of K940 as collective variables in order to monitor local side-chain motions (as demonstrated in *Figure 4*). The drug was not included in the collective variables because it is diffusive. In total, the collective variables contain 1659 atoms, consisting of 4,977 Cartesian coordinates.

String method calculations were performed with NAMD (*Phillips et al., 2005*) combined with in-house scripts. After defining 30 images along the initial pathway from the targeted MD trajectory, atomistic structure around each atomic structure was relaxed by 35 ns equilibration imposing positional harmonic restraints on the collective variables to its image (*Figure 2—figure supplement 2*). After the relaxation, the mean forces type string method (*Maragliano et al., 2006*) was conducted by using positional harmonic restraints with a force constant of 0.1 kcal/Å$^2$. Mean forces were evaluated every 4 ns of simulation and images were updated according to the calculated mean force, then reparameterized to make the images equidistant from each other (also a weak smoothing operation was applied). In order to eliminate external components (translations and rotations) in the mean forces, the Cα atoms of TM helices in snapshots were fitted to the reference structure (*Branduardi and Faraldo-Gómez, 2013*) before mean force estimation. The reference structure was created by averaging the crystal structures of the BEA, EAB (120 degrees rotation about the z-axis from BEA) and ABE states (240 degrees rotation), thus resulting in a 3-fold symmetric structure (*Zhu and Hummer, 2010*). The least-squares fittings of structures were performed only in the xy-plane. z-coordinates were kept the same as in the original snapshots because the free energies of membrane proteins are symmetric only in the xy-plane. The terminal images (images 1 and 30) were kept fixed during the first 36 ns (nine cycles in the string method), and then allowed to move to relax subtle frustrations in the crystal structure that might result from detergent molecules or crystal packing. In order to avoid any drifts of the terminal images along the pathway, tangential components to the pathway were eliminated from the mean force when updating terminal images (*Zhu and Hummer, 2010*). Convergences of the images were monitored by RMSDs of images from those of the initial pathway (*Figure 2—figure supplement 3*). Also, we confirmed that terminal images converged to the same regions as those sampled by brute-force simulations (*Figure 2—figure supplement 4*).

In order to obtain more statistics and to evaluate free energy profiles along pathways, umbrella samplings were carried out using a single umbrella window per image. Umbrella samplings, each of 30 ns length, were performed around the images of the initial pathway, then over 30 ns around the images after the 9th cycle in the string method, over 10 ns after the 14th cycle, over 15 ns after the

19th cycle, and over 100 ns after the 24th cycle (the final converged pathway). The positional harmonic restraints with a rather weak force constant of 0.01 kcal/Å$^2$ were used to achieve sufficient phase space overlaps between adjacent systems. Trajectory data were post-processed by MBAR (*Matsunaga et al., 2012*; *Shirts and Chodera, 2008*) and weights for the restraint-free condition were obtained. Free-energy profiles along the pathways were evaluated using the progress coordinate (*Branduardi et al., 2007*; *Matsunaga et al., 2016*) (which measures a tangential component) for pathways. To focus on the samples within a reactive tube (*Ren and Vanden-Eijnden, 2005*), distance metric (*Branduardi et al., 2007*) (which measures an orthogonal component) from pathway was introduced, and samples that deviated further than a cutoff radius of 2,000 Å$^2$ were ignored. Changing the cutoff radius to 1500 or 3000 Å$^2$ did not alter the results qualitatively. Statistical uncertainties (standard errors) in the free-energy profiles were estimated from block averages. For analysis of the converged pathway (*Figures 2B, C* and *5*), only samples from those images were used in order to eliminate any biases arising from the pathways of earlier cycles.

For the simulations with NAMD under NPT ensemble (300 K and 1 atm), Langevin dynamics and Nose-Hoover Langevin piston (*Feller et al., 1995*; *Martyna et al., 1994*) were used. Long-range electrostatic interactions were calculated using PME with a grid size of <1 Å. The Lennard-Jones interactions were treated with the force switch algorithm (over 8–10 Å). SHAKE (*Ryckaert et al., 1977*) and SETTLE (*Miyamoto and Kollman, 1992*) were applied with a time step of 2 fs. A multiple time-stepping integration scheme was used with a quintic polynomial splitting function to define local force components. Calculations were performed by a hybrid OpenMP/MPI scheme on the K computer of RIKEN AICS (*Yokokawa et al., 2011*).

## Alchemical free energy calculations

In order to quantify contributions from the drug–protein interactions to the free energy profile along the pathway, we conducted alchemical free energy calculations (*Klimovich et al., 2015*) (double-decoupling method) with NAMD and we evaluated the absolute drug-binding energy for each image of system 2. The last snapshots of the minimum free energy pathway of system 2 were used for the initial structure. In the calculation, minocycline was gradually decoupled with the other atoms by using a coupling parameter λ with 25 intermediate λ-states (stratifications). The simulation length for each λ-state was 1 ns. During the decoupling simulations, the same positional harmonic restraints (force constant of 0.1 kcal/Å$^2$) as were used for the string method were imposed on the collective variables. No restraints were imposed on minocycline because it was hard to define binding sites for images after the shrinkage of the distal binding site. Free energy differences over the decoupling were evaluated by the exponential averages (*Zwanzig, 1955*) of the differences of the potential energies between adjacent λ states. Statistical uncertainties were estimated from block averages. The same type of decoupling simulation was performed for minocycline solvated by TIP3P water molecules. The water box size comparable to the AcrB system was used to cancel out the system size effect (*Hummer et al., 1996*). From two free energy differences, an absolute binding energy was calculated for each image by considering a thermodynamic cycle. The error propagation rule was applied to obtain final statistical uncertainties. To reduce computation time, absolute binding energies were calculated only for images 1, 4, 5, 7, 10–23 and 28–30.

In addition, to evaluate the impact of protonation of D408 in either the B state or the E state, we conducted another type of alchemical free energy calculation. Imposing positional harmonic restraints (force constant of 0.1 kcal/Å$^2$) on the collective variables of image 5 of system 1 (D408p in protomer I, and D408 in protomer II), the protonation states were alchemically transformed to that of system 2 (D408 in protomer I, and D408p in protomer II). Because of the restraints on the collective variables, this calculation mainly evaluates the contribution of protonation to the free energy differences. Transformations were carried out in both forward and backward directions, with 36 intermediate λ-states for each direction, resulting in a total simulation length of 58 ns. During the transformation, the total charge of the system was kept neutral. Free energy difference of the alchemical transformation was evaluated by BAR (*Bennett, 1976*) (implemented in the ParseFEP toolkit [*Liu et al., 2012*]) and statistical uncertainty was estimated analytically. The same type of alchemical transformation was also conducted starting from the collective variables of image 15 of system 2 to the protonation state of system 1.

In order to obtain more statistics before and after the alchemical transformation, we conducted MD simulations of 50 ns length for the initial and final state, imposing the same type of positional harmonic restraints as above. Then, for the investigation of electrostatic potential energy maps, snapshots taken from these simulations were analyzed by the *k*-space Gaussian split Ewald method (*Shan et al., 2005*), and instantaneous electrostatic potentials of the smooth part were interpolated and averaged over snapshots on the fixed 3D grids. Most of the analyses were performed with in-house MATLAB scripts (publicly available at https://github.com/ymatsunaga/mdtoolbox under the BSD 3-Clause License) (*Matsunaga, 2018*) and MDTraj (*McGibbon et al., 2015* [copy archived at https://github.com/elifesciences-publications/mdtoolbox]). Molecular figures were generated with VMD (*Humphrey et al., 1996*).

### Mutual information analysis

While the minimum free-energy path obtained by the string method provides us with the physically most probable pathway in the collective variable space, it is not useful in characterizing the dynamic properties of conformational change at the atomistic level (e.g., atoms not involved in the collective variables). Here, in order to characterize the dynamic properties of atomistic fluctuations, we generated reactive trajectories from the umbrella sampling data around the minimum free-energy path. The reactive trajectories are defined as the portions of trajectories when, after leaving the initial BEA state, it enters first the final EBA state before returning to the BEA state (*E and Vanden-Eijnden, 2010*). For generating the reactive trajectories, we used the property that the minimum free-energy pathway is orthogonal to the isocommittor surface (*Maragliano et al., 2006*). It is known that the distribution of the reactive trajectories follows the equilibrium distribution restricted on the isocommittor surface (*Ren and Vanden-Eijnden, 2005*). Using this property, we resampled the reactive trajectories from the umbrella sampling data according to the weights calculated by the MBAR in the previous analysis. Under discretization by 29 slices orthogonal to the pathway, 100 reactive trajectories were resampled from the umbrella sampling data. Then, mutual information of reactive trajectories was computed for pairs of residues, one from the TM and the other from the porter domains by,

$$MI = \int \int p(r_1, r_2) \ln \frac{p(r_1, r_2)}{p(r_1), p(r_2)} dr_1 dr_2,$$

where Cartesian coordinates of the center-of-masses of sidechains were used for $r_1$ and $r_2$ because they better capture correlated motions involving semi-rigid regions (*McClendon et al., 2014*; *Vanwart et al., 2012*) than did the dihedral angles often used in this kind of analysis (*McClendon et al., 2009*). $p(r_1)$ and $p(r_1, r_2)$ are a probability density of $r_1$, and a joint probability density of $r_1$ and $r_2$, respectively. Unlike usual correlation coefficients, the mutual information contains information about both linear and nonlinear dependences. For numerical evaluation of the mutual information, Kraskov's algorithm (*Kraskov et al., 2004*), based on *k*-nearest neighbor distances (without binning), was used because it is practical for multi-dimensional data sets such as Cartesian coordinate data.

## Acknowledgements

We acknowledge help from Tomio Kamada and Naoyuki Miyashita in running NAMD on the K computer. We also would like to thank Yuji Sugita, Chigusa Kobayashi, Jaewoon Jung, Motoshi Kamiya, Suyong Re, Yohei Koyama, Shun Sakuraba, Luca Maragliano, and Shoji Takada for beneficial discussions. Computational resources were provided by HOKUSAI in RIKEN, and K computer in RIKEN Advanced Institute for Computational Science by the HPCI System Research project (Project ID: hp120027). This research was partly supported by Research and Development of the Next-Generation Integrated Simulation of Living Matter, by RIKEN Advanced Institute for Computational Science (to YM), by KAKENHI (Grant Numbers 24770159 [to YM] and 25291036 [to MI]), by JST PRESTO (Grant No. JPMJPR1679 to YM), by the Platform Project for Supporting Drug Discovery and Life Science Research from AMED (to KM, MI and AK), by Innovative Drug Discovery Infrastructure through Functional Control of Biomolecular Systems, by Priority Issue one in Post-K Supercomputer Development from MEXT (Project IDs hp150269, hp160223 and hp170255, to TY and MI), and by the RIKEN Dynamic Structural Biology Project (to MI).

# Additional information

## Funding

| Funder | Grant reference number | Author |
| --- | --- | --- |
| Japan Society for the Promotion of Science | 24770159 | Yasuhiro Matsunaga<br>Mitsunori Ikeguchi |
| Japan Science and Technology Agency | JPMJPR1679 | Yasuhiro Matsunaga |
| Research Organization for Information Science and Technology | hp120027 | Yasuhiro Matsunaga<br>Tohru Terada<br>Kei Moritsugu<br>Hiroshi Fujisaki<br>Mitsunori Ikeguchi<br>Akinori Kidera |
| Japan Society for the Promotion of Science | 25291036 | Yasuhiro Matsunaga<br>Mitsunori Ikeguchi |
| Ministry of Education, Culture, Sports, Science, and Technology | hp150269 | Tsutomu Yamane<br>Mitsunori Ikeguchi |
| Ministry of Education, Culture, Sports, Science and Technology | hp160223 | Tsutomu Yamane<br>Mitsunori Ikeguchi |
| Ministry of Education, Culture, Sports, Science and Technology | hp170255 | Tsutomu Yamane<br>Mitsunori Ikeguchi |
| Japan Agency for Medical Research and Development | Platform Project for Supporting Drug Discovery and Life Science Research | Kei Moritsugu<br>Mitsunori Ikeguchi<br>Akinori Kidera |
| RIKEN Dynamic Structural Biology Project | | Mitsunori Ikeguchi |
| RIKEN Advanced Institute for Computational Science | | Yasuhiro Matsunaga |

The funders had no role in study design, data collection and interpretation, or the decision to submit the work for publication.

## Author contributions

Yasuhiro Matsunaga, Conceptualization, Data curation, Software, Formal analysis, Funding acquisition, Validation, Investigation, Visualization, Methodology, Writing—original draft, Writing—review and editing; Tsutomu Yamane, Conceptualization, Data curation, Software, Formal analysis, Funding acquisition, Investigation, Methodology, Writing—original draft, Writing—review and editing; Tohru Terada, Kei Moritsugu, Conceptualization, Software, Investigation, Methodology, Writing—review and editing; Hiroshi Fujisaki, Conceptualization, Investigation, Methodology, Writing—review and editing; Satoshi Murakami, Conceptualization, Data curation, Supervision, Writing—original draft, Writing—review and editing; Mitsunori Ikeguchi, Conceptualization, Data curation, Software, Supervision, Funding acquisition, Methodology, Writing—original draft, Writing—review and editing; Akinori Kidera, Conceptualization, Resources, Data curation, Supervision, Funding acquisition, Methodology, Project administration, Writing—review and editing

## Author ORCIDs

Yasuhiro Matsunaga http://orcid.org/0000-0003-2872-3908
Tohru Terada http://orcid.org/0000-0002-7091-0646

## Decision letter and Author response

Decision letter https://doi.org/10.7554/eLife.31715.021
Author response https://doi.org/10.7554/eLife.31715.022

## Additional files

**Supplementary files**

• Transparent reporting form
DOI: https://doi.org/10.7554/eLife.31715.019

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
