## [Decision Letter]

Thank you for submitting your article "Energetics and Conformational Pathways of Functional Rotation in the Multidrug Transporter AcrB" for consideration by *eLife*. Your article has been favorably evaluated by Richard Aldrich (Senior Editor) and two reviewers, one of whom, Yibing Shan (Reviewer #1), is a member of our Board of Reviewing Editors.

The reviewers have discussed the reviews with one another and the Reviewing Editor has drafted this decision to help you prepare a revised submission.

Summary:

Aiming to elucidate the detailed structural mechanism of AcrB RND transporter, this study applied the string method for conformational free-energy calculations on a supercomputing platform to identify the minimum free energy pathway of the functional rotation of AcrB. The all-atom simulations follow the conformational change of the transporter including the travel of the drug molecule (minocycline) from the binding pocket through the gate and finally release of the drug molecule, showing that the protonation of Asp408 in a transmembrane helix of the Binding protomer drives the process. This work represents potentially important progress in our understanding of the molecular mechanism of the transport process in AcrB.

Major points:

Given the readership of *eLife*, the manuscript should include a brief and high level description of the simulation techniques (the string method, the umbrella sampling, and the alchemic free energy calculation) in the main text in terms that a reader unfamiliar with molecular dynamics simulations can relate to. What are the basic ideas and premises behind the approach? The manuscript should explain the free energy reported in Figure 2, stating clearly that this is the free energy for the whole system if that's the case.

The specific procedure should also be explained in more detail as is it is difficult to follow. It would also help if the references to the string method in the last paragraph of the Introduction were made in a more structured way – with explanations, rather than the current lumping together of a large number of very diverse papers ranging from mathematical principles all the way to practical applications.

This work only attempts to distinguish two possible schemes. In realty the process of the functional rotation is probably more complicated, possibly involving protonation of not only Asp408. For instance, Asp924 is located in a quite hydrophobic local environment and its protonation is conceivable. The proton could even be passed from Asp408 to Asp924 by the single water file connecting them. Ideally some new simulations should be performed to address this. At the very least this possibility should be more carefully discussed.

The explanation for the increase in free energy after the minima (Figure 2) invoking drug release without uptaking new drug molecule without offering any reasoning or evidence seems casual. The reviewers suggest that the apparent high free energy might be due to the fact that any change of protonation state is not accounted for in the simulations. The authors should consider deprotonation of Asp408, which should occur at the end of the extrusion phase according to the proposed scheme (Figure 6). The authors should also consider, as aforementioned, protonation of other residues (e.g. Asp924) following Asp408 deprotonation (a single water file seems to connect Asp408 and Asp924).

Why is the drug released from the BEA complex in System 1, if the protonation of the original E protomer stabilizes the whole complex, as shown in the Figure 5 of Figure 2? How is the BEA configuration changed to EAB after protonation of the second protomer as suggested by Figure 1?

The analysis concerning the dynamics coupling of the transmembrane domains and the porter and the funnel domains (Figure 5) is too preliminary and very unintuitive. Some abstract metric of share conformation may show that the coupling is present, which is hardly surprising, but biologically it is much more interesting to discuss/demonstrate what interactions are behind the coupling.

---

## [Author Response]

Major points:Given the readership of eLife, the manuscript should include a brief and high level description of the simulation techniques (the string method, the umbrella sampling, and the alchemic free energy calculation) in the main text in terms that a reader unfamiliar with molecular dynamics simulations can relate to. What are the basic ideas and premises behind the approach? The manuscript should explain the free energy reported in Figure 2, stating clearly that this is the free energy for the whole system if that's the case.The specific procedure should also be explained in more detail as is it is difficult to follow. It would also help if the references to the string method in the last paragraph of the Introduction were made in a more structured way – with explanations, rather than the current lumping together of a large number of very diverse papers ranging from mathematical principles all the way to practical applications.

Following the reviewers’ comment, we added the descriptions of the basic ideas behind the simulation techniques and their specific procedures in the Introduction section. The references to the string method are now explained in a more structured way.

This work only attempts to distinguish two possible schemes. In realty the process of the functional rotation is probably more complicated, possibly involving protonation of not only Asp408. For instance, Asp924 is located in a quite hydrophobic local environment and its protonation is conceivable. The proton could even be passed from Asp408 to Asp924 by the single water file connecting them. Ideally some new simulations should be performed to address this. At the very least this possibility should be more carefully discussed.The explanation for the increase in free energy after the minima (Figure 2) invoking drug release without uptaking new drug molecule without offering any reasoning or evidence seems casual. The reviewers suggest that the apparent high free energy might be due to the fact that any change of protonation state is not accounted for in the simulations. The authors should consider deprotonation of Asp408, which should occur at the end of the extrusion phase according to the proposed scheme (Figure 6). The authors should also consider, as aforementioned, protonation of other residues (e.g. Asp924) following Asp408 deprotonation (a single water file seems to connect Asp408 and Asp924).

We appreciate the reviewers’ comment on this point. We agree with the reviewers that other protonation schemes (e.g., Asp924) should be considered. Actually, proton translocation occurs from the periplasm (Figure 1 top, the side of Asp924) to the cytoplasm (Figure 1 bottom) in the secondary transporter, there is a possibility that protons are passed from Asp924 to Asp408 at the initial state (proton uptake event in the BEA complex). Therefore, we simulated a protonated Asp924 of protomer I at the initial state (Figure 1) for both system 1 and system 2 in this revision, as well as another possibility of protonation at Glu346. Glu346 was chosen because the pKa calculations by Eicher et al., 2014 suggested protonation at this residue. Results were added in Figure 3—figure supplement 2. The electrostatic potential map after the protonation at Asp924 shows a local positive change around Asp924, which may stabilize the system by the interactions with the oxygens of the phosphate groups of lipids, and help to recruit water molecules for proton uptake. However, the impact of the protonation at Asp924 is spatially localized compared to that of Asp408, and the change in the electrostatic potential was not propagated to the transmembrane region. The protonation at Glu346 scarcely changed the electrostatic potential. Considering these results, we believe that the protonation of these sites do not change the story of our study on the energetics of the large-scale conformational change of the whole protein. We added the discussion in the first paragraph of the subsection “Protonation induces a vertical shear motion in the TM domain”.

As for the increase in free energy at the end state (the EAB complex), we would like to point out that the end state of system 2 (the EAB complex with protonated Asp408 in protomer I) is identical to the initial state of system 1 (the BEA complex with protonated Asp408 in protomer II) by a 120° rotation, if we ignore the drug binding/unbinding. Since our free energy calculations already showed that the initial state of system 1 is stable with the bound drug (shown in Figure 2), we consider that the drug unbinding is the main cause for the unstability of the end state of system 2. We added the discussion in the second paragraph of the subsection “Energetics of functional rotation under two different protonation states”.

Why is the drug released from the BEA complex in System 1, if the protonation of the original E protomer stabilizes the whole complex, as shown in the Figure 5 of Figure 2? How is the BEA configuration changed to EAB after protonation of the second protomer as suggested by Figure 1?

The methods were not clearly explained. Before optimizing the pathway with the string method, we generated the initial guess for the conformational transition pathway using targeted molecular dynamics from the initial structure (the BEA complex) toward the end structure (the EAB complex). During the simulation, the drug was also forced toward the exit gate. In response to the first comment, we briefly explained the procedure also in the main text, including how we created the initial pathway.

The analysis concerning the dynamics coupling of the transmembrane domains and the porter and the funnel domains (Figure 5) is too preliminary and very unintuitive. Some abstract metric of share conformation may show that the coupling is present, which is hardly surprising, but biologically it is much more interesting to discuss/demonstrate what interactions are behind the coupling.

The description on the dynamical coupling is presented in Figure 4 and 5. First, Figure 4 exhibits a coupling between the TM domain and the porter domain. Then, in Figure 5, to identify the interactions behind the coupling, we presented the results of the exhaustive correlation analysis (based on mutual information) for all residue pairs of the porter and TM domains. The figure reveals that the coil-to-helix transition of TM8 (caused by the sidechain flip of Phe459) is the very interaction causing the coupling, and regulates the closing motion of the porter subdomains, PN1/PC2, which extrudes the drug. We believe that the coil-to-helix transition of TM8 transmits the events occurred in the TM domain to the porter domain, and produces the biological function, the extrusion of a drug.

We agree with the reviewers that mutual information is not necessarily intuitive compared to other metrics such as linear correlation or distance. However, the protein contains different types of motions, the closing rotational motion of the porter domain and the translational downward motion of the TM helices. It was thus not possible to capture such complicated nonlinear motions with the linear metrics. Mutual information is one of the major metrics to treat nonlinear relations. More or less, other nonlinear metrics are also nonintuitive. Thus, we have tried to improve the readability of the text by changing the way of the presentation instead of changing the analysis method. Since the description of the mutual information given in Figure 5 is too detailed, we moved it to a new Figure 5 supplement. Instead, we moved Figure 1—figure supplement 1 to new Figure 5, which compares the crystal structures of TM8 for different states as well as the sidechain positions of Phe459. Now, the readers can readily understand how the coil-to-helix transition of TM8 occurs. According to the updates of the figures, the main text was modified.